# EstablishINg the best STEp-up treatments for children with uncontrolled asthma despite INhaled corticosteroids (EINSTEIN): protocol for a systematic review, network meta-analysis and cost-effectiveness analysis using individual participant data (IPD)

Sofia Cividini [1], Ian Sinha [2], Sarah Donegan [1], Michelle Maden [3], Giovanna Culeddu [4], Katie Rose [2], Olive Fulton,[5] Dyfrig A Hughes [4], Stephen Turner [6], Catrin Tudur Smith [1]

IS and CTS are joint senior authors.

For numbered affiliations see end of article.

**Correspondence to**
Professor Catrin Tudur Smith;
cat1@liverpool.ac.uk

## ABSTRACT

**Introduction** Asthma affects millions of children worldwide—1.1 million children in the UK. Asthma symptoms cannot be cured but can be controlled with low-dose inhaled corticosteroids (ICSs) in the majority of individuals. Treatment with a low-dose ICS, however, fails to control asthma symptoms in around 10%–15% of children and this places the individual at increased risk for an asthma attack. At present, there is no clear preferred treatment option for a child whose asthma is not controlled by low-dose ICS and international guidelines currently recommend at least three treatment options. Herein, we propose a systematic review and individual participant data network meta-analysis (IPD-NMA) aiming to synthesise all available published and unpublished evidence from randomised controlled trials (RCTs) to establish the clinical effectiveness of pharmacological treatments in children and adolescents with uncontrolled asthma on ICS and help to make evidence-informed treatment choices. This will be used to parameterise a Markov-based economic model to assess the cost-effectiveness of alternative treatment options in order to inform decisions in the context of drug formularies and clinical guidelines.

**Methods and analysis** We will search in MEDLINE, the Cochrane Library, the Cochrane Central Register of Controlled Trials (CENTRAL), Embase, NICE Technology Appraisals and the National Institute for Health Research (NIHR) Health Technology Assessment series for RCTs of interventions in patients with uncontrolled asthma on ICS. All studies where children and adolescents were eligible for inclusion will be considered, and authors or sponsors will be contacted to request IPD on patients aged <18. The reference lists of existing clinical guidelines, along with included studies and relevant reviews, will be checked to identify further relevant studies. Unpublished studies will be located by searching across a range of clinical trial registries, including internal trial registers for

## Strengths and limitations of this study

► This study will be the most comprehensive synthesis of interventional studies of children and adolescents with uncontrolled asthma on inhaled corticosteroid.

► It will be able to establish the most clinically and cost-effective step-up therapy for asthma, also considering those treatments where no head-to-head comparison exists, providing evidence to make informed treatment choices.

► It will be able to evaluate the influence of patient-level characteristics on treatment effect.

► It will provide an assessment of the incremental cost per quality-adjusted life-year gained of treatment options, from the perspective of the National Health Service and personal social services.

► Individual participant data may not be available from all eligible randomised controlled trials and integration with aggregate data may be needed.

pharmaceutical companies. All studies will be appraised for inclusion against predefined inclusion and exclusion criteria by two independent reviewers with disagreements resolved through discussion with a third reviewer. We will perform an IPD-NMA—eventually supplemented with aggregate data for the RCTs without IPD—to establish both the probability that a treatment is best and the probability that a particular treatment is most likely to be effective for a specific profile of the patient. The IPD-NMA will be performed for each outcome variable within a Bayesian framework, using the WinBUGS software. Also, potential patient-level characteristics that may modify treatment effects will be explored, which represents one of the strengths of this study.

**Ethics and dissemination** The Committee on Research Ethics, University of Liverpool, has confirmed that ethics

review is not required. The dissemination plan consists of publishing the results in an open-access medical journal, a plain-language summary available for parents and children, dissemination via local, national and international meetings and conferences and the press offices of our Higher Education Institutions (HEIs). A synopsis of results will be disseminated to NICE and British Thoracic Society/Scottish Intercollegiate Guidelines Network (SIGN) as highly relevant to future clinical guideline updates.

**PROSPERO registration number** CRD42019127599.

## INTRODUCTION
### Rationale

Recent epidemiological data show that an always-increasing number of people of all age groups worldwide, among that millions of children and adolescents, are affected by asthma with a notable social and economic burden for health systems.[1] Asthma remains a common medical condition affecting over one million children in the UK, one of the highest prevalence rates worldwide.[2 3] Asthma is characterised by symptoms of wheeze, breathlessness, chest tightness and cough and can affect the child's quality of life (QoL) by limiting daily activities and causing acute attacks. In the UK, it was estimated that one child is admitted to the hospital every 20 min because of an asthma attack.[2]

The British guideline[4 5] on the management of asthma recommends that, following a diagnosis of asthma in a child, a stepwise approach to treatment should be taken. The initial level of treatment is a low-dose inhaled corticosteroid (ICS) to prevent symptoms plus a short acting $\beta_2$-adrenoceptor agonist (SABA) to relieve symptoms. However, treatment with a low-dose ICS fails to control asthma symptoms in around 10%–15% of children.[6] If asthma remains uncontrolled, a series of further steps are followed consisting of a treatment step-up by including add-on preventer therapies such as long-acting $\beta_2$-adrenoceptor agonists (LABAs) or leukotriene receptor antagonists (LTRAs), increasing the dose of ICS or adding sustained-release (SR) theophylline.[1 4 5]

Choosing the best step-up treatment thus becomes a crucial decision to prevent exacerbations and to avoid poor asthma control, also improving the QoL of patients and their families and optimising the use of National Health Service (NHS) resources.[7–12] At present, no clear preferential option for initial step-up exists, and the decision-making process about treatment choices is mainly based on clinicians' preferences and not on evidence-based practice. Moreover, there is substantial heterogeneity among individuals in the treatment response.[13 14]

A recent Cochrane review[15] comprising of 33 trials with 6381 children demonstrated that adding LABAs to ICSs was not associated with a significant decrease in the rate of exacerbations requiring systemic steroids but was superior for improving lung function (eg, forced expiratory volume in 1 s ($FEV_1$) and peak expiratory flow (PEF)), compared with the same or higher doses of ICS. No statistically significant group difference in hospital admissions was found. Notwithstanding, because of a trend towards

increased risk of hospital admission with LABA independently of ICS dose, the authors advised further monitoring of treatment.[15]

Another Cochrane review[16] published in 2013 found that adding LABAs to ICSs was associated with no statistically significant reduction neither of the use of oral corticosteroids (OCS) as rescue nor hospital admissions compared with the same or an increased dose of ICS. Evidence was based only on four trials with 559 children and adolescents having mild-to-moderate asthma.[16] The authors cautioned that the lack of paediatric trials, the absence of data on preschoolers and the variability in the reporting of relevant clinical outcomes was a limit for firm conclusions regarding this comparison.[16]

Two network meta-analyses (NMAs), analysing published randomised controlled trial (RCT) aggregate-level data for children with uncontrolled asthma, have already been conducted,[17 18] but the evidence from these analyses has important limitations. The considerable variation in and the incomplete reporting of outcome measurements across RCTs prevented a formal NMA and assessment of relative efficacies of treatments in one NMA of 23 RCTs (4129 patients).[18] In 2015, Zhao et al[17] conducted a formal NMA consisting of 35 RCTs with 12010 children. These authors suggested that combined ICS and LABA treatments were most effective in preventing exacerbations and that treatments with medium-dose or high-dose ICS, combined ICS/LTRA and low-dose ICS seem to be equally effective.[17] However, the analysis has limitations as the authors excluded 70 relevant RCTs because publications of these RCTs did not provide data about 'exacerbations' or 'symptom-free days'. Outcome reporting bias[19] is, therefore, a severe threat to the validity of their results if the excluded studies had selectively reported results based on the statistical significance of their findings. Moreover, a series of other issues make the interpretation and generalisability of the results of this NMA difficult. There is a lack of complete overlapping of the included studies compared with the previous Cochrane reviews[15 16] and previous planned NMA[18]—only seven RCTs are in common across the two NMAs.[17 18] Two other weak points are represented by the lack of analyses focussing on outcomes of importance and relevance to patients and the lack of analyses comparing different drugs (ICS, LABA, LTRA), doses and types of inhalation devices within ICS and LABA classes. We have also included tiotropium in the search algorithm as a long-acting muscarinic antagonists (LAMA), but we are not confident of finding sufficient studies.

Several RCTs have compared two alternative step-up options head to head, but only a few individual trials have analysed more than two classes head to head. The Best Add-on Therapy Giving Effective Responses (BADGER) trial[14] randomised 182 children and adolescents with uncontrolled asthma on fluticasone to receive each of the following therapies: 'ICS step-up', 'LABA step-up' and 'LTRA step-up', obtaining differential responses to every step-up treatment. 'LABA step-up' showed

significantly more likely to provide the 'best' response; yet, many patients better replied with 'ICS or LTRA step-up' suggesting that for some patients, these treatments may be preferred.[14] Overall, results from BADGER trial were heterogeneous and a 16-week follow-up may not be sufficient.

The Management of Asthma in School age Children On Therapy (MASCOT) trial[20] randomised children aged 6–14 to fluticasone plus placebo, fluticasone/salmeterol plus placebo and fluticasone plus montelukast. However, this RCT failed to recruit an adequate number of patients due to significant organisational challenges, issues of accessing patients from primary care and changes in prescribing habits during the trial. These challenges limit the potential success of similar future clinical trials and recommendations have been made for the use of alternative study designs[20 21] to compare alternative step-up treatments.

There is thus an urgent need for a novel approach to be taken to synthesise all of the available evidence using robust, unbiased methods, also including unpublished data. A key strength of the EstablishINg the best STEp-up treatments for children with uncontrolled asthma despite INhaled corticosteroids (EINSTEIN) analysis will be the assessment of individual patient-level characteristics in modifying the patient's response to treatment. No study has previously considered this aspect, although it represents a crucial clinical step in evaluating treatment efficacy. Furthermore, an economic analysis of the treatment options will provide evidence on their cost-effectiveness from the perspective of the NHS in the UK. Results from the EINSTEIN Study will provide clinicians and patients with accessible, high-quality, patient-relevant information to help make evidence-informed treatment choices. Earlier identification of the best step-up treatment for a particular child could have a significant impact on children's lives with more extensive benefits to society and the NHS.

## Objectives

Our research objective is to establish the best treatment option for patients younger than 18 whose asthma symptoms are uncontrolled by low–medium dose ICS. To this goal, the EINSTEIN Study will identify and synthesise all available evidence from RCTs to establish the effectiveness of pharmacological treatments in children and adolescents aged <18 having poor asthma control on ICS. The study will also investigate the modifiers of treatment effect (eg, age, ethnicity, asthmatic phenotype) to optimise a targeted therapy and maximise patients' informed choice of treatment. The economic analysis will estimate the cost-effectiveness of treatments in terms of their incremental costs per quality-adjusted life-year (QALY) gained.

Specifically, we aim to:
1. Conduct a systematic review to identify relevant RCTs of treatment with children and adolescents whose asthma remains uncontrolled despite ICS use.
2. Collect individual participant data (IPD) from all eligible trials.
3. Carry out an NMA of IPD to identify the most effective treatment.
4. Identify modifiers of treatment effect to establish which patients respond better to each treatment.
5. Develop an economic model to estimate the incremental cost per QALY gained of treatment options, from the perspective of the NHS and Personal Social Services.
6. Identify remaining uncertainties formulating recommendations to inform priorities aimed at future research.
7. Disseminate findings.

## METHODS

The Preferred Reporting Items for Systematic Reviews and Meta-Analysis Protocols (PRISMA-P) guidelines[22] have guided the drafting of this protocol.

### Eligibility criteria

For inclusion in the EINSTEIN Study, we will select studies according to the criteria outlined below.

### Study designs

We will include parallel and cross-over RCTs, of any duration, using any level of blinding and comparing at least one of the health technologies of interest. All RCTs meeting our inclusion criteria will be included irrespective of the outcomes reported in the publication, to reduce the potential for outcome reporting bias.

### Participants

We will focus on children (<12 years) and adolescents (12–17 years) of any ethnicity with poor asthma control on ICS before randomisation as defined by the study protocol. We will include studies with mixed-age groups in which children and adolescents were eligible because we will contact authors concerning specific data on patients aged <18. Patients aged ≥18, or with stable asthma as defined by the study authors, or using an asthma medication other than ICS, or an ICS with an add-on active (eg, ICS/LABA) as treatment regimen before randomisation will not be considered. Participants using any dose ICS at randomisation will be eligible for inclusion.

### Interventions

We will consider as eligible all studies where children and adolescents were randomised to at least one of the following treatments:
► ICSs, alone or in combination with other treatments—beclometasone dipropionate; ciclesonide; fluticasone propionate; fluticasone furoate; budesonide and mometasone.
► LABAs—formoterol; salmeterol and vilanterol.
► LTRAs—zafirlukast and montelukast.
► Theophylline.

Any dose of preventer treatment—oral or inhaled, and any inhaler devices (pressurised metered-dose inhaler, dry powder inhaler, combination inhaler)—will be considered. Presence or absence of spacer devices will be noted.

We will not consider studies with participants randomised to any asthma medication other than those listed in the inclusion criteria.

## Comparators

For both primary and secondary outcomes—if not specified otherwise—patient outcomes will be compared at two levels, namely among:

1. Every class of treatment—ICS (alone or with add-on actives), LABA, LTRA, theophylline and placebo. For ICS, we will distinguish between low, medium and high doses as defined in Global Initiative for Asthma (GINA)[11]—(table 1 and figure 1).
2. Every drugs within every class of treatment—beclometasone dipropionate; ciclesonide; fluticasone propionate; fluticasone furoate; budesonide; mometasone; formoterol; salmeterol; vilanterol; zafirlukast; montelukast; theophylline and placebo.

Moreover, we will explore hierarchical models that consider both classes of treatments and actives with interactions of covariates. Healthy controls will not be considered as an appropriate reference group.

## Outcomes

Review outcomes will include the outcomes identified as important to healthcare practitioners and patients during the development of a core outcome set for trials in children with asthma.[23 24]

Primary outcomes
► Exacerbations as defined by European Respiratory Society (ERS)/American Thoracic Society (ATS).[24]
► Asthma control measured by a validated test (eg, asthma control test (ACT),[25] asthma control questionnaire (ACQ),[26–29] other).

Secondary outcomes
► Symptoms/symptom score.
► QoL.
► Mortality—although rare, this is an important outcome for parents.
► Physiological outcomes (eg, $FEV_1$ and bronchial responsiveness).
► Adverse effects (AEs)—including growth and withdrawals due to AEs.
► Hospital admissions.
► Total healthcare costs (obtained by the sum–product of item of resource use and their unit cost), resource use and utility outcomes to inform the economic model.

## Timing

Studies will be selected for inclusion independently of the length of follow-up of outcomes.

## Setting

There will be no restrictions by type of setting.

**Table 1** Estimated clinical comparability daily doses (µg) of inhaled corticosteroids

### ≤5-year old (children)

| Drug | Low dose | Medium dose | High dose |
|---|---|---|---|
| Beclomethasone dipropionate (HFA) | 100 (≥5 years) | N.A. | N.A. |
| Budesonide nebulised | 500 (≥1 year) | N.A. | N.A. |
| Budesonide pMDI+spacer | N.A. | N.A. | N.A. |
| Fluticasone propionate (HFA) | 50 (≥4 years) | N.A. | N.A. |
| Mometasone furoate | 110 (≥4 years) | N.A. | N.A. |
| Ciclesonide | N.A. | N.A. | N.A. |

### 6–11-year old (children)

| Drug | Low dose | Medium dose | High dose |
|---|---|---|---|
| Beclomethasone dipropionate (CFC) | 100–200 | >200–400 | >400 |
| Beclomethasone dipropionate (HFA) | 50–100 | >100–200 | >200 |
| Budesonide (DPI) | 100–200 | >200–400 | >400 |
| Budesonide (nebules) | 250–500 | >500–1000 | >1000 |
| Ciclesonide | 80 | >80–160 | >160 |
| Fluticasone furoate (DPI) | N.A. | N.A. | N.A. |
| Fluticasone propionate (DPI) | 100–200 | >200–400 | >400 |
| Fluticasone propionate (HFA) | 100–200 | >200–500 | >500 |
| Mometasone furoate | 110 | ≥220-<440 | ≥440 |

### ≥12-year old (adults and adolescents)

| Drug | Low dose | Medium dose | High dose |
|---|---|---|---|
| Beclomethasone dipropionate (CFC) | 200–500 | >500–1000 | >1000 |
| Beclomethasone dipropionate (HFA) | 100–200 | >200–400 | >400 |
| Budesonide (DPI) | 200–400 | >400–800 | >800 |
| Ciclesonide (HFA) | 80–160 | >160–320 | >320 |
| Fluticasone furoate (DPI) | 100 | N.A. | 200 |
| Fluticasone propionate (DPI) | 100–250 | >250–500 | >500 |
| Fluticasone propionate (HFA) | 100–250 | >250–500 | >500 |
| Mometasone furoate | 110–220 | >220–440 | >440 |

CFC, chlorofluorocarbon propellant (no longer used; included for comparison with older literature); DPI, dry powder inhaler; HFA, hydrofluoroalkane propellant; N.A, not applicable; pMDI, pressurized metered dose inhaler.

## ≤ 5-year-old (Children)

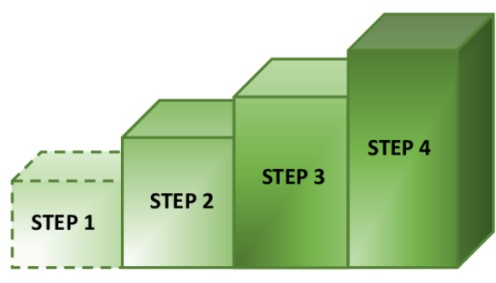

**STEP 1** SABA as needed
**STEP 2** Daily low dose ICS – SABA as needed.
*Other controller options*: LTRA, or intermittent ICS.
**STEP 3** Double low dose ICS – SABA as needed
*Other controller options*:  low dose ICS + LTRA.
**STEP 4** Continue controller + specialist assessment
*Other controller options*: add LTRA / increase ICS
frequency / add intermittent ICS.

Reliever: as-needed SABA

## 6-11-year-old (Children)

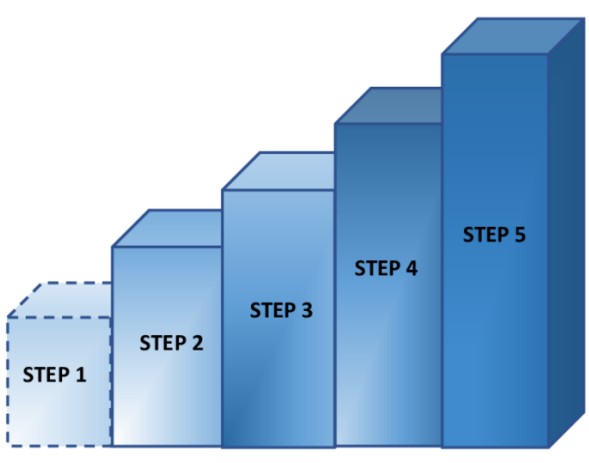

**STEP 1** SABA as needed.
*Other controller options*: low dose of ICS taken with
SABA, or daily low dose ICS.
**STEP 2** Daily low dose ICS – SABA as needed.
*Other controller options*: LTRA, or low dose ICS taken
with SABA.
**STEP 3** Low dose ICS+LABA, or medium dose ICS –
SABA as needed. *Other controller options*: low dose
ICS+LTRA.
**STEP 4** Medium dose ICS+LABA – SABA as needed.
*Other controller options*: high dose ICS+LABA, or add-on
tiotropium, or add-on LTRA.
**STEP 5** Phenotypic assessment ± add-on therapy (e.g.,
anti-IgE) – SABA as needed. *Other controller options*:
add-on anti-IL5, or add-on low dose OCS.

Reliever: as-needed SABA

## ≥ 12-year-old (Adults and adolescents)

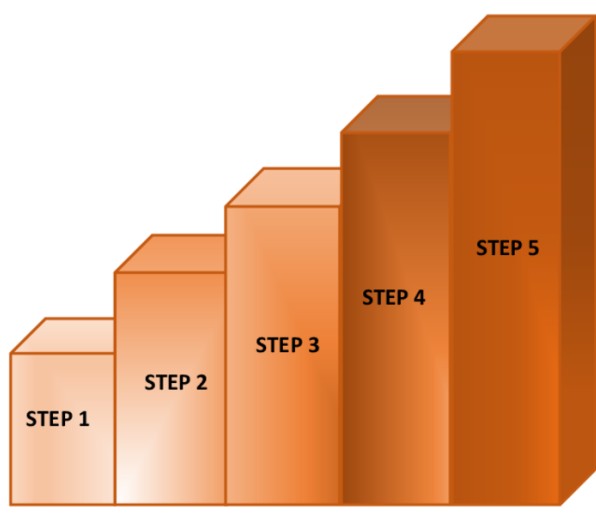

**STEP 1** Low dose ICS/formoterol as needed.
*Other controller options*: low dose of ICS taken with
SABA.
**STEP 2** Daily low dose ICS – Low dose ICS/formoterol as
needed. *Other controller options*: LTRA, or low dose ICS
taken with SABA.
**STEP 3** Low dose ICS+LABA – Low dose ICS/Formoterol
as needed. *Other controller options*: medium dose ICS, or
low dose ICS+LTRA.
**STEP 4** Medium dose ICS+LABA – Low dose
ICS/Formoterol as needed. *Other controller options*: high
dose ICS, add-on tiotropium, or add-on LTRA.
**STEP 5** High dose ICS+LABA. Phenotypic assessment ±
add-on therapy (e.g., tiotropium, anti-IgE, anti-IL5/5R, anti-
IL4R) – Low dose ICS/Formoterol as needed.
*Other controller options*: add-on low dose OCS.

Preferred reliever: as-needed low dose ICS/formoterol
Reliever option: as-needed SABA

**Figure 1** Treatment step-up for asthmatic patients according to age group. Adapted from Global Initiative for Asthma.[1] ICS,
inhaled corticosteroid; IL, interleukin; LABA, long-acting $\beta_2$-adrenoceptor agonist; LTRA, leukotriene receptor antagonist; OCS,
oral corticosteroids; SABA, short-acting $\beta._2$-adrenoceptor agonist.

## Language

We will include only articles reported in the English language.

## Information sources and search strategy

Our search strategy will identify published and unpublished studies and builds on previously published search strategies for aggregate data NMAs[17 18] and Cochrane reviews[15 16 30–32] (see online supplemental file 1; for example, MEDLINE search). The search strategies will run from 2014 onwards only. From inception to 2013, we will consider as exhaustive the searches already carried out for the meta-analyses and Cochrane reviews mentioned above,[15–18 30–32] and we will retrieve all the corresponding studies.

We will search MEDLINE, the Cochrane Library, the Cochrane Central Register of Controlled Trials, Embase, National Institute for Health and Care Excellence (NICE) Technology Appraisals and the NIHR Health Technology Assessment series using relevant search terms. We will also scan the reference list of included studies and relevant reviews along with the reference lists of existing clinical guidelines such as the British Thoracic Society (BTS) guideline[4 5] and GINA.[1]

For unpublished studies, we will search a range of clinical trial registries enclosed inside the WHO 'International Clinical Trials Registry Platform' search portal (including clinicaltrials.gov and International Standard Randomised Controlled Trial Number (ISRCTN)) and conference abstracts (eg, European Respiratory Society and American Thoracic Society).

We will also search internal clinical trial registers for pharmaceutical companies that manufacture health technologies of interest (eg, GSK, AstraZeneca, Novartis, Merck).

We will limit the search to English language articles, and we will verify that RCTs have included participants aged <18 as a subset.

## Study records
### Data management

Studies identified for screening of abstracts and full-text articles will be managed within Covidence (https://www.covidence.org/home), a systematic review management tool.

### Selection process

All studies will be screened and assessed for inclusion by two independent reviewers (SC and KR) with disagreements resolved through discussion with a third reviewer (IS). To minimise the impact of selective outcome reporting, we will include all RCTs meeting our inclusion criteria irrespective of the outcome(s) reported in the publication. The authors will thus be contacted to ask for information and the IPD for all outcomes relevant to the review.

### Data collection process

The first author, or sponsor, of each included trial will be approached and asked to supply anonymised IPD, metadata and relevant documentation[33] (protocol and blank case report forms) from the respective trial, either through direct contact or through data sharing portals such as clinicalstudydatarequest.com, where relevant.

Before starting the quantitative synthesis, we will conduct a set of standard quality and consistency checks on the provided data. This will involve crosschecks by reanalysing IPD against previously published results to highlight inconsistencies or possible errors. Any queries will be raised with the original trialists wherever possible. Data will be cleaned and standardised to allow pooling and subsequent analyses of the data.

## Data items

The data requested from each clinical trial will include at least:

► Baseline characteristics—age; sex; ethnicity; eczema; height; weight; baseline severity; baseline PEF rate (PEFR) and $FEV_1$. Where available, we will also collect other variables such as age at asthma diagnosis; asthma-related hospitalisation and exacerbation history; allergic sensitisation; parental asthma and objective indicators of asthma such as bronchodilator reversibility or methacholine PC20.

► The date of randomisation and dates of follow-up visits/interval between randomisation and follow-up.

► Treatment details including inhalation device and dose.

► Adherence data if available.

► Data for the review outcomes along with details of their definitions and measurement tools used—symptoms; exacerbations; asthma control; mortality; QoL; growth; physiological outcomes: PEFR and $FEV_1$ and hospital admissions.

► Cost; resource use and utility outcomes to inform the economic model.

We will use data from the intent-to-treat (ITT) population.

## Outcomes and prioritisation
### Primary outcomes

The primary outcomes of the study are exacerbations and loss of/change in asthma control measured by a validated test.

1. Exacerbations are defined as in ERS/ATS,[24] namely '*events characterised by a change from the patient's previous status*'. We will define exacerbations as every event requiring the use of OCS, the need for unscheduled visits with general practitioners (GPs) or at the emergency department (ED) and hospitalisation.

2. Based on the definition reported in ERS/ATS,[24] asthma control corresponds to '*the extent to which the various manifestations of asthma have been reduced or removed by treatment*'.

Exacerbations are expected to be an individual-level numerical variable (binary, ordinal or count) or an individual-level time-to-event variable. Asthma control is expected to be an individual-level numerical variable

**Table 2** A possible categorisation of asthma control outcome based on ACT/ACQ Scores

| Test | Asthma control | Total score |
|---|---|---|
| ACT 4–11 (years) | 0=poor control 1=good/total control | Score≤19 Score=20–27 |
| ACT 12+ (years) | 0=poor control 1=good/total control | Score≤19 Score=20–25 |
| ACQ | 0=poor control 1=good/total control | Score>1 Score≤1 |
| Others | 0=poor control 1=good/total control | To be evaluated on an individual case by case basis |

Note: if both ACT and ACQ are present, we will choose ACT as the best choice.
ACQ, asthma control questionnaire; ACT, asthma control test.

(categorical) and will be measured by a validated test (eg, ACT,[25] ACQ,[26–29] etc). Table 2 represents a possible categorisation for asthma control outcome based on ACT/ACQ Scores. A standardisation method coherently to ACT and ACQ will be applied for other kinds of tests. After consultation with clinicians, we will decide if, for this outcome variable, it is worthier to consider the full longitudinal profile (multiple measurements) or to establish a clinically relevant timepoint.

### Secondary outcomes
The secondary outcomes of the study are as follows.
1. Symptoms/symptom score: The mode to report symptoms or symptom score may be different across studies. We will distinguish between daytime and night-time symptoms, whatever they are, also considering their intensity.
2. QoL: It may have been measured differently across the studies. We will use the most frequent scale as a model to transform the remaining data using, for example, established conversion tables.
3. Mortality: This variable may be reported in different ways across studies. It will always be converted to a binary variable.
4. Physiological measurements: $FEV_1$ in L/s and PEFR%. After consulting clinicians, we will decide if considering the full longitudinal profile (multiple measures) or establishing a clinically relevant timepoint.
5. Adverse events: These may differ across the studies. We will consider relevant AEs, such as cardiac and neuropsychiatric disorders, infections, pneumonia, behavioural difficulty, growth and diabetes.
6. Hospital admissions: This variable may be reported in different ways across studies. We will look for evidence such as dates of hospitalisation, adverse events indicating admission occurred and variables directly reporting hospitalisation, between the randomisation visit and the end of the follow-up.

### Economic outcomes
7. The primary economic outcome will be the incremental cost per QALY gained. This will be estimated from data on:
   a. Hospital admissions.
   b. Other items: resource use, costs, and utilities—these will be based on purposive searches of the economic and medical literature.

### Risk of bias in individual studies
Two independent reviewers will use the Cochrane risk of bias tool[34] to record risk of bias concerning randomisation method, allocation concealment, blinding, incomplete outcome data and selective reporting. Disagreements will be resolved through discussion with a third reviewer.

### Data
*Synthesis*
For every separate outcome, we will create a NMA diagram displaying the number of studies and patients for each treatment comparison within the network.

We will fit a Bayesian hierarchical meta-analysis model—supplemented with aggregate data if necessary—to synthesise the available IPD. We will thus estimate the relative treatment effect—odds ratio (OR) for categorical data and difference in means for continuous data—and credibility interval for every pairwise comparison based on direct evidence. We will assess the homogeneity assumption by comparing the deviance information criterion (DIC) of fixed-effect and random-effect models and observing the between-trial variance. The forest plots, $\chi^2$ test for heterogeneity and $I^2$ statistic will be examined to assess the evidence of heterogeneity within each pairwise meta-analysis based on direct evidence.

We will conduct an NMA for every outcome within a Bayesian framework with the WinBUGS software with the goodness of fit assessed by calculating the posterior mean residual deviance with DIC used as a basis for model comparison. Correlation between treatment effects from multiarm trials will be appropriately accounted for. From the NMA, we can estimate the relative treatment effect for every pairwise comparison irrespective of they have been compared directly in an RCT as well as calculating the probability that each treatment is the best or worst for a given outcome. In random effects NMA models, we will check the conventional assumption that between-trial variance is the same for each comparison, by fitting pairwise models based on direct evidence and assessing whether the variance is similar for each comparison. If the assumption appears unrealistic, we will explore other variance structures for the NMA model.

In an NMA, the indirect comparisons are not protected by randomisation, as for direct comparisons. So, they may be affected by confounding due to differences among clinical trials.[35] The validity of an NMA thus depends on the critical assumption of transitivity, namely to verify whether the probability of having been able to have given any of the treatment in the network to any patient in the

network was equally likely.[35] We will check this assumption by comparing the inclusion/exclusion criteria of trials to make a judgement about whether patients, trial protocols, doses and administration procedures are similar in the modes that might modify treatment effect. We will use model fit and selection statistics to informally assess whether inconsistency is evident along with a formal analysis using a 'node-splitting' approach.[36 37]

We will explore potential patient-level characteristics that may modify treatment effects using hierarchical models with interaction effects between treatment and specific covariates. Initially, based on direct evidence and subsequently in an NMA of the IPD, with aggregate data if IPD is unavailable.[38–41] The effect of covariates will be separated within and between trials. The underlying consistency assumption of these models will also be explored. Patient-level characteristics of interest include age, gender, ethnicity, eczema status, asthmatic phenotype, baseline PEFR% and asthma severity. A literature search will be conducted to identify any further potential characteristics to explore.

We will fit an NMA model to compare compounds within each class and separate models to compare the different classes of treatments. Furthermore, complex hierarchical models that account for both classes and compounds with covariate interactions will be explored.

We will undertake sensitivity analyses for investigating the impact of missing IPD at the level of results and conclusions of NMA and for exploring the robustness of results to different priors in the Bayesian analyses. If sufficient data are available, sensitivity analyses that adjust for levels of adherence will be conducted to assess the robustness of results obtained from the primary analysis on the ITT patients.

### Metabias(es)

To minimise the impact of selective outcome reporting, we will include all RCTs meeting our inclusion criteria irrespective of the outcome(s) reported in the publication. For analyses containing 10 or more trials—due to the low power of the assessments for analyses containing small numbers of trials—we will investigate publication bias (and other selection bias/small study effects) by using funnel plots and appropriate statistical tests for funnel plot asymmetry.

### Confidence in cumulative evidence

We will assess the quality of evidence for all outcomes across the domains of risk of bias, consistency, directness, precision and publication bias according to Grading of Recommendations Assessment, Development and Evaluation (Grade).

### Economic analysis

The economic analysis will be conducted using standard methods,[42] based on the most robust data available and reported according to the Consolidated Health Economic Evaluation Reporting Standards (CHEERS) statement.[43]

Cost-effectiveness will be estimated from the perspective of the NHS and Personal Social Services in line with NICE guidance[44] and based on an economic model that considers health outcomes, resource use, costs and health utilities to estimate the incremental cost per QALY gained of each treatment. This analysis will not consider missed school or work.

The parameters needed to populate the models will include outcomes estimated by the NMA of the IPD (eg, efficacy and safety parameters) and other parameters (eg, utilities, resource use and the long-term costs of care) that will require searching of evidence beyond the studies included in the NMA. These will be sourced from a purposive review of the literature and by using specialist databases (eg, NHS Economic Evaluation Database (EED) for resource use parameters and the Tufts CEA registry for utility parameters) where necessary. Unit cost data will be derived from standard national sources (eg, NHS reference costs[45] and the British National Formulary[46]).

A Markov model will be necessary for longer-term extrapolation of costs and outcomes, to allow for any differential impacts of treatments over time and for transition of patients among the five steps of treatment.[47] Results of the NMA will be converted to transition probabilities based on standard methods.[48 49] The model will evaluate costs and outcomes over the lifetime of the patient cohorts: costs and benefits in future years will be discounted at an annual rate of 3.5% and varied between 0% and 6% in sensitivity analysis. Results from the model will be reported as incremental cost per QALY gained incremental cost-effectiveness ratios (ICERs) and compared with the NICE threshold range of £20 000–£30 000 per QALY.

Uncertainties in all parameter inputs will be accounted for in the analysis by including parametric distributions for each point estimate, supplemented by expert opinion, where the evidence is not sufficiently detailed. This will enable probabilistic sensitivity analyses to be performed based on sampling from distributions using Monte Carlo simulation. The NMA estimates relative effects jointly, and the full joint distribution of these effects will be used in the economic model in order to preserve correlation.[50] Uncertainty in the optimally cost-effective treatment will be represented by cost-effectiveness acceptability curves (CEACs),[51] which present the probability that each drug is the most cost-effective at a given threshold of cost-effectiveness. Where the IPD-NMA indicates that patient characteristics modify treatment effects, a stratified approach will be used to assess cost-effectiveness in particular patient subgroup(s). Standard techniques (eg, extreme value scenarios) will be used to ensure the internal validity of the model.

We will conduct a value of information analysis to inform future research priorities, for example, whether more short-term efficacy trials are needed, or more long-term follow-up, or more data on the utilities, or costs to reduce decision uncertainty.[52] The expected value of perfect information (EVPI) and the expected value of

perfect parameter information (EVPPI) will be calculated on both per-patient and population levels using the Sheffield accelerated value of information approximation[53] to facilitate computation effort.

## ETHICS AND DISSEMINATION

The Committee on Research Ethics, University of Liverpool, has confirmed that ethics review is not required. The dissemination plan consists of publishing the results in an open-access medical journal, a plain-language summary available for parents and children, dissemination via local, national and international meetings and conferences and the press offices of our HEIs. A synopsis of results will be disseminated to NICE and BTS/SIGN as highly relevant to future clinical guideline updates.

## Patient and public involvement

The EINSTEIN protocol has been developed in consultation with children with asthma and their parents and with NHS clinicians who routinely care for children with uncontrolled asthma in NHS settings. First, we have sought advice on our proposal and the lay summary from five families, including two children, who attended our asthma clinic at Alder Hey. Second, the outcomes chosen in our review have been selected from a core outcome set which were agreed as important by clinicians and patients.[21] This earlier work was heavily influenced by patients and, in particular, highlighted the importance to parents of including mortality and long-term AEs as outcomes. Finally, we consulted an Alder Hey's patient advisory group, comprising children with asthma and their parents and asked them to comment specifically on the lay summary and choice of outcomes in the review project. We will consult with the patient group towards the end of the project to help prepare a patient-friendly podcast or leaflet that could be shared on asthma websites and in clinics. We will share our findings at important international meetings and in medical journals, and we will make sure that this process reflects issues that are important to children and families. Our patient representative (OF) will help to coordinate the group of patients and parents, along with physicians involved. Patients will not be involved in the recruitment to and conduct of the study.

**Author affiliations**
[1]Department of Health Data Science, University of Liverpool, Liverpool, UK
[2]Alder Hey Children's Foundation NHS Trust, Liverpool, UK
[3]Liverpool Reviews and Implementation Group (LRIG), University of Liverpool, Liverpool, UK
[4]Centre for Health Economics and Medicines Evaluation, Bangor University, Bangor, UK
[5]Patient Representative, Liverpool, UK
[6]Department of Child Health, University Court of the University of Aberdeen, Aberdeen, UK

**Correction notice** This article has been corrected since it first published. The provenance and peer review statement has been included.

**Acknowledgements** The authors would like to thank patients and their families for their contribution to this project.

**Contributors** CT-S is the guarantor. CT-S and IS conceived the study. CT-S, SD, ST, DAH, OF and IS drafted the original version of the protocol, and SC drafted this manuscript. All authors contributed to the development of the selection and data extraction criteria and the strategy of risk of bias assessment. MM developed the search strategy. CT-S and SD provided statistical expertise. IS and ST provided clinical expertise. KR and SC screened and selected eligible studies for inclusion in the review. DAH and GC developed and wrote the plan for the health economics analysis. OF contributed to coordinating the group of patients and parents and developing the plain-language summary. All authors read, provided feedback and approved the final manuscript.

**Funding** This work was supported by NIHR Health Technology Assessment Programme under grant number 16/110/16. The funder had no role in the protocol development. The views expressed are those of the authors and not necessarily those of the National Health Service, the NIHR or the Department of Health.

**Competing interests** None declared.

**Patient consent for publication** Not required.

**Provenance and peer review** Not commissioned; externally peer reviewed.

**ORCID iDs**
Sofia Cividini http://orcid.org/0000-0003-2705-9224
Ian Sinha http://orcid.org/0000-0002-7342-5523
Sarah Donegan http://orcid.org/0000-0003-1709-2290
Michelle Maden http://orcid.org/0000-0003-4419-6343
Giovanna Culeddu http://orcid.org/0000-0001-5032-4255
Katie Rose http://orcid.org/0000-0002-2348-2036
Dyfrig A Hughes http://orcid.org/0000-0001-8247-7459
Stephen Turner http://orcid.org/0000-0001-8393-5060
Catrin Tudur Smith http://orcid.org/0000-0003-3051-1445

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
