## [Reviewer comments · BMJ Open]

ARTICLE DETAILS

TITLE (PROVISIONAL)	EstablishING the best STEp-up treatments for children with uncontrolled asthma despite INhaled corticosteroids (EINSTEIN): Protocol for a systematic review, network meta-analysis and cost-effectiveness analysis using individual participant data (IPD)
AUTHORS	Cividini, Sofia; Sinha, Ian; Donegan, Sarah; Maden, Michelle; Culeddu, Giovanna; Rose, Katie; Fulton, Olive; Hughes, Dyfrig; Turner, Stephen; Tudur-Smith, Catrin

VERSION 1 – REVIEW

REVIEWER	Salman Mroueh American University of Beirut, Lebanon
REVIEW RETURNED	10-Jul-2020

GENERAL COMMENTS	This research once completed should offer some evidence-based guidance on the management of asthma in children, a very prevalent condition. I am concerned however about the feasibility of the economic analysis: how could the authors compare different outcomes of research carried out in countries with different health care systems, over different periods, using different medications (various formulations of inhaled steroids, more or less expensive...)?
---

REVIEWER	David Mauger Penn State University, USA
REVIEW RETURNED	26-Oct-2020

GENERAL COMMENTS	Overall this is a well-conceived NMA protocol. My comments relate mainly to the need for additional information or minor corrections. Lines 136/215 Are you intentionally excluding long-acting anti-muscarinics (LAMA) such as tiotropium? Also, in several places your writing seems to indicate that LTRAs are inhaled drugs. They are not. If you exclude trials that included non-inhaled formulations (line 215), there will not be any LTRA trials in your NMA. Line 197 - It is not clear as to whether the criteria for poor control apply at the trial level or at the participant level. That is, will a trial be excluded if some, but not all, participants do not have poor asthma control? If it is at the participant level, this would seem to go against the use of the full ITT population. Line 201 - You should clarify what you mean by "entering" the trial. I'm guessing you mean the point of randomization as opposed to the point of entry into the pre-randomization phase of the study. If you do mean the point of entry into the pre-randomization phase of
--

	the study, this criteria may cause complications. Line 248 - I understand how resource use and utility outcomes are defined, but I'm not sure exactly what you mean by "cost." Line 297/384 - Other baseline characteristics to potentially include might be: age at asthma diagnosis, asthma related hospitalization and exacerbation history, allergic sensitization, parental asthma, objective indicators of asthma such as bronchodilator reversibility or methacholine PC20. Line 313-You may want to consider other recent definition of asthma exacerbation (eg, Fuhlbrigge et al, JACI 2012 v129(3 suppl)) in which systemic corticosteroids are required. Unscheduled GP visits without systemic corticosteroids may represent quite mild events. Line 340 - Other economic outcomes to potentially include might be missed school or lost work. Line 350 (Major comment) - Plans for data harmonization are lacking. You jump right into the statistical analysis plan. How will you harmonize different measure of control such as ACT and ACQ? Will they be treated as different outcomes, in which case you may run into disjointed analyses, or will they be harmonized into a single measure of control through some form of standardization. How will harmonize symptom reporting? Will exacerbations be treated as time-to-event or frequency outcomes? How will your harmonize outcomes assessed at different time points? Line 353 (Major comment) - How, exactly, will the model be "supplemented with aggregate data?" Line 366 - Is there some threshold for "best?" What if one treatment increases lung function by 10% and another by 10.2%, is the first better than the second or is there some zone of equivalence? Line 376 (Major comment) - It doesn't make sense to examine this by comparing only the inclusion/exclusion criteria of the trials. You will need to look at other aspects too. In addition, will the judgement be subjective or objective? If objective, what will be the criteria? Line 382 - You mention in the introduction that patient level characteristics may be very important in this clinical context. How will you incorporate patient level characteristics in aggregate data? Line 396 (Major comment) - How will you integrate analyses for different outcomes that are done using different trials? You mention, quite rightly, that you will not exclude from the project any trials based on the absence of any particular outcome. However, some trials will be excluded (by definition) from analyses on outcomes that are not reported for that trial. How will you deal with that potentially source of bias? Line 420 - I have not seen Markov models employed in the context of NMA. Are there references for this? If not, could you add a bit more detail as to how you will accomplish it when different trials have different amounts of follow-up? Figure 1 - It would be worth double-checking your Figure against GINA. For example, you have "Low dose ICS/Formoterol as needed"
--	--

	as therapies at both step 3 and step 4 in the >12 group. Also, I don't think any of the controllers are recommended to be given "as-needed" at the higher steps. I think they are recommended to be given daily.
--	--

VERSION 1 – AUTHOR RESPONSE

Reviewer: 1
Reviewer Name

Salman Mroueh

Institution and Country

American University of Beirut, Lebanon

Please state any competing interests or state 'None declared':
None declared

Comments to the Author

This research once completed should offer some evidence-based guidance on the management of asthma in children, a very prevalent condition.

I am concerned however about the feasibility of the economic analysis: how could the authors compare different outcomes of research carried out in countries with different health care systems, over different periods, using different medications (various formulations of inhaled steroids, more or less expensive...)?

We thank Dr Mroueh for the time dedicated to the revision of our manuscript and the valuable comments provided. Please, our answers below.

Response: In relation to the economic analysis,

(i) Our original manuscript specified that our perspective was from the National Health Service and Personal Social Services in the UK, and therefore a single healthcare system. For example lines 79, 182 and 410.

(ii) We will implement a multi-state model in which patients will transition or move to another health state at any point in time. Transitions among these health states are determined by the effectiveness of the different treatments being assessed. The assumption of generalisability of clinical effectiveness is no different to that of cost-effectiveness in this respect.

(iii) UK unit costs will be attached to items of resource use (which will include primary care, hospitalisation, medicines), and the total cost calculated for each health state. It is the different rates of transition of patients among these health states (because of treatments having differential effectiveness) that determines the different total costs of treatments.

(iv) We will use NHS costs of specific asthma treatment products.

(v) Generalisability assumption – this applies only to clinical effectiveness and health-related quality of life (utilities). Resource use and costs will be specific to the UK.

These points are included in the manuscript, or implicit in standard methods of economic evaluation. Suggest no change necessary.

Reviewer: 2
Reviewer Name

David Mauger

Institution and Country

Penn State University, USA

Please state any competing interests or state 'None declared':

None

Comments to the Author

Overall this is a well-conceived NMA protocol. My comments relate mainly to the need for additional information or minor corrections.

We thank Dr Mauger for his time and helpful comments. Please, our answers below.

Lines 136/215 Are you intentionally excluding long-active anti-muscarinics (LAMA) such as tiotropium?

Also, in several places your writing seems to indicate that LTRAs are inhaled drugs. They are not. If you exclude trials that included non-inhaled formulations (line 215), there will not be any LTRA trials in your NMA.

Many thanks for this comment. We have clarified that LTRA are included and that these are not inhaled medication. We have clarified that we included tiotropium in the search algorithm, but that we are not confident of finding enough studies (Line 138, marked-up manuscript).

Line 197 - It is not clear as to whether the criteria for poor control apply at the trial level or at the participant level. That is, will a trial be excluded if some, but not all, participants do not have poor asthma control? If it is at the participant level, this would seem to go against the use of the full ITT population.

Many thanks for highlighting this. Poor control refers to the trial level, according to the definition used by the original trial protocol as one of the inclusion criteria of the EINSTEIN study. No patient will be excluded because of having poor asthma control during the treatment phase after randomisation, coherently with the definition of the ITT population. We have clarified this (Line 201, marked-up manuscript).

Line 201 - You should clarify what you mean by "entering" the trial. I'm guessing you mean the point of randomization as opposed to the point of entry into the pre-randomization phase of the study. If you do mean the point of entry into the pre-randomization phase of the study, this criteria may cause complications.

We have clarified that we mean randomisation (Lines 201 and 207, marked-up manuscript).

Line 248 - I understand how resource use and utility outcomes are defined, but I'm not sure exactly what you mean by "cost."

Many thanks for this comment. We have changes this to "Total health care costs (obtained by the sum-product of item of resource use and their unit cost)" (Lines 255-72).

Line 297/384 - Other baseline characteristics to potentially include might be: age at asthma diagnosis, asthma related hospitalization and exacerbation history, allergic sensitization, parental asthma, objective indicators of asthma such as bronchodilator reversibility or methacholine PC20.

Many thanks. In the protocol, we have added these helpful suggestions to the protocol (Lines 308-310 marked version).

Line 313-You may want to consider other recent definition of asthma exacerbation (eg, Fuhlbrigge et al, JACI 2012 v129(3 suppl)) in which systemic corticosteroids are required. Unscheduled GP visits without systemic corticosteroids may represent quite mild events.

Many thanks. We believe our approach (i.e., including GP visits with/without oral steroids and hospital admissions) will give insight into treatment effect on severity of exacerbations. No modification made.

Line 340 - Other economic outcomes to potentially include might be missed school or lost work.

Many thanks for this comment. In line 424 of the original manuscript (line 431 of the marked version) we stated: "The cost-effective analysis will adopt the perspective of the NHS and Personal Social Services in line with NICE guidance (reference 44)". Societal perspective costs, such as inability to work or missing school will therefore not be part of this analysis. We have added a short sentence: "This analysis will not consider missed school or work." (Line 434).

Line 350 (Major comment) - Plans for data harmonization are lacking. You jump right into the statistical analysis plan. How will you harmonize different measure of control such as ACT and ACQ? Will they be treated as different outcomes, in which case you may run into disjointed analyses, or will they be harmonized into a single measure of control through some form of standardization. How will harmonize symptom reporting? Will exacerbations be treated as time-to-event or frequency outcomes? How will your harmonize outcomes assessed at different time points?

Many thanks for this suggestion. We have now added a new paragraph (Lines 331-338 marked version) and a table (Table 2) to address this point.

Line 353 (Major comment) - How, exactly, will the model be "supplemented with aggregate data?"

Each model has two (FE model) or three (RE model) parts. Part 1 models IPD for the first trial set, part 2 models AD for the second trial set, and part 3 models trial-specific treatment effects, allowing all trials to contribute to the estimation of the treatment effect and between-trial variance [Donegan S, Williamson P, D'Alessandro U, et al. Combining individual patient data and aggregate data in mixed treatment comparison meta-analysis: Individual patient data may be beneficial if only for a subset of trials, Stat Med 2013;32(6):914–30].

We will consider models without interactions and with treatment-by-covariate interactions, and we will account for correlation among multi-arm trials. No modification made.

Line 366 - Is there some threshold for "best?" What if one treatment increases lung function by 10% and another by 10.2%, is the first better than the second or is there some zone of equivalence?

For each NMA the probability that each treatment in the network is ranked best, second best, third best etc will be calculated using Monte Carlo simulations and compared across different outcomes using graphical summaries and surface area under the cumulative ranking (SUCRA) which will be interpreted alongside the pairwise effect estimates and 95% credibility intervals (Salanti G, Ades AE, Ioannidis JPA. Graphical methods and numeric summaries for presenting results for multiple-treatment meta-analysis: an overview and tutorial. J Clin Epidemiol. 2011;64(2):163–171. doi: 10.1016/j.jclinepi.2010.03.016). No modification made.

Line 376 (Major comment) - It doesn't make sense to examine this by comparing only the inclusion/exclusion criteria of the trials. You will need to look at other aspects too. In addition, will the judgement be subjective or objective? If objective, what will be the criteria?

Many thanks for this consideration. The comparison between the inclusion/exclusion criteria of the trials is the first step, and it should be the most objective possible (Line 379).

Formally, we will use the node-splitting approach too (Line 380). In Bayesian hierarchical models, the dependency structure can be represented with a directed acyclic graph (DAG). In the DAG, parameters to estimate correspond to nodes. Node-based methods to detect a conflict between different information sources on a specific parameter exist. These methods — known as a node-splitting approach — are based on splitting different sources of information at the level of a node and comparing the resulting posterior distributions. This comparison allows identifying any conflicts between these different sources of information. In a NMA, the node-splitting approach is used to split into the 'direct' component and the 'indirect' component the information contributing to the estimates of a relative effect parameter d_{XY} (node). For the mean treatment effect d_{XY} , it is possible to obtain a posterior distribution based on studies comparing the treatments X and Y directly with mean d_{XY}^{dir} , and a posterior distribution based on a NMA of all the remaining studies with mean d_{XY}^{ind} . Evaluating if the

hypothesis that the posterior distributions from direct and indirect evidence are in agreement can be supported by data through the examination of the following index:

$$\omega_{XY} = d_{XY}^{dir} - d_{XY}^{ind}$$

ω_{XY} represents the posterior distribution of the inconsistency parameter. It is possible to obtain a measure of conflict P defined as the probability that ω_{XY} is greater than 0.

The full NMA model and the model with a particular split node can be compared using the posterior mean of the residual deviance and the DIC. A reduction in the posterior mean of the residual deviance or the DIC represents a signal of a possible inconsistency between the different evidence sources for that node.

No modification made.

Line 382 - You mention in the introduction that patient level characteristics may be very important in this clinical context. How will you incorporate patient level characteristics in aggregate data?

Many thanks for this observation. Of course, it is not possible to do that with aggregate data. However, the integration with aggregate data — if IPD is not available for a subset of clinical trials — will not affect the value of results, as already shown in previous studies.

- Riley RD, Lambert PC, Staessen JA, et al. Meta-analysis of continuous outcomes combining individual patient data and aggregate data. Stat Med 2008;27:1870–93.*
- Donegan S, Williamson P, D'Alessandro U, et al. Combining individual patient data and aggregate data in mixed treatment comparison meta-analysis: Individual patient data may be beneficial if only for a subset of trials. Stat Med 2013;32(6):914–30.].*

No modification made.

Line 396 (Major comment) - How will you integrate analyses for different outcomes that are done using different trials? You mention, quite rightly, that you will not exclude from the project any trials based on the absence of any particular outcome. However, some trials will be excluded (by definition) from analyses on outcomes that are not reported for that trial. How will you deal with that potentially source of bias?

Thanks for this comment. We will not exclude "a priori" publications where the outcome(s) reported do not match the EINSTEIN outcomes. Conversely, we will contact the authors enquiring whether they have collected one or more of the outcomes of our interest, although not reported in the publication. If so, we will ask them for data. If not so, the RCT will be excluded because it does not meet our needs. This issue cannot be considered as a source of bias as independent of the procedure of data collection. Yet, we will take note of that in the discussion of the results.

No modification made.

Line 420 - I have not seen Markov models employed in the context of NMA. Are there references for this? If not, could you add a bit more detail as to how you will accomplish it when different trials have different amounts of follow-up?

Thanks for this comment. There are several examples of Markov model transition probabilities being calculated from the results of NMA (one example of a well-conducted study is: doi: 10.3310/hta21090). RCTs which present different cycle lengths or follow-up or that report results in different ways (e.g. some might report information on number of events, number of patients experiencing at least one event, and total number of events, etc) can all be statistically combined by exploiting mathematical relationships between the Markov transition parameters (See Welton N.J., Sutton A.J., Cooper N.J., Abrams K.R., Ades, A.E. 2012. Evidence Synthesis for Decision Making in Healthcare, John Wiley & Sons, Ltd., and Gidwani R, Russell LB. Estimating Transition Probabilities from Published Evidence: A Tutorial for Decision Modelers. Pharmacoeconomics. 2020

Nov;38(11):1153-1164). We have now referred to these text in the manuscript.

Figure 1 - It would be worth double-checking your Figure against GINA. For example, you have “Low dose ICS/Formoterol as needed” as therapies at both step 3 and step 4 in the >12 group. Also, I don’t think any of the controllers are recommended to be given “as-needed” at the higher steps. I think they are recommended to be given daily.

Thanks for the suggestion. We have rechecked Figure 1, and we have slightly modified it. All relievers are reported as as-needed. You can check for that in the figures on pages 46, 47, and 143 (GINA-2019-main-report-June-2019-wms).

VERSION 2 – REVIEW

REVIEWER	David Mauger Penn State University USA
REVIEW RETURNED	02-Dec-2020
GENERAL COMMENTS	The reviewer completed the checklist but made no further comments.